# LC-MS/MS-Based Fungicide Accumulation Assay to Demonstrate Efflux Activity in the Wheat Pathogen *Zymoseptoria tritici*

**DOI:** 10.3390/microorganisms10081494

**Published:** 2022-07-25

**Authors:** Guillaume Fouché, Dominique Rosati, Catherine Venet, Hervé Josserand, Marie-Pascale Latorse, Danièle Debieu, Sabine Fillinger

**Affiliations:** 1UR BIOGER, INRAE, Université Paris-Saclay, 78850 Thiverval-Grignon, France; guillaume.fouche182@gmail.com (G.F.); daniele.debieu@club-internet.fr (D.D.); 2La Dargoire Research Center, Bayer SAS, 69009 Lyon, France; drosati@hotmail.fr (D.R.); catherine.venet@bayer.com (C.V.); herve.josserand@bayer.com (H.J.); marie-pascale.latorse@bayer.com (M.-P.L.)

**Keywords:** multidrug resistance, intracellular drug accumulation, medium-throughput assay, non-radioactive assay

## Abstract

Increased drug efflux compromises the efficacy of a large panel of treatments in the clinic against cancer or bacterial, fungal, and viral diseases, and in agriculture due to the emergence of multidrug-resistant pathogenic fungi. Until recently, to demonstrate increased drug efflux, the use of labeled drugs or fluorescent dyes was necessary. With the increasing sensitivity of detection devices, direct assessment of drug efflux has become realistic. Here, we describe a medium-throughput method to assess the intracellular drug concentration in the plant pathogenic fungus *Zymoseptoria tritici* cultivated in the presence of a sublethal fungicide concentration. As a model fungicide, we used the succinate-dehydrogenase inhibitor boscalid. The boscalid concentration was assessed in the different culture fractions using mass spectrometry linked to liquid chromatography (LC-MS/MS). The ratio between the intracellular and total boscalid amount was used as an inversed proxy for the efflux activity. Using isogenic mutant strains known for their differential efflux capacities, we validated the negative correlation between the intracellular boscalid concentration and efflux activity. In addition, intra-cellular fungicide accumulation explains the susceptibility of the tested strains to boscalid. This assay may be useful in lead development when a new molecule displays good inhibitory activity against its isolated target protein but fails to control the target organism.

## 1. Introduction

The adaptation of drug-targeted organisms through resistance development is an issue encountered in both the clinical and agricultural sectors [1,2]. It occurs in almost all concerned organisms, namely fungi, yeasts, bacteria, viruses, insects, parasites, weeds, and cancer cells [3]. Resistance may result from a large variety of mechanisms, among which one can distinguish specific mechanisms that will affect one particular drug (or several drugs with the same mode of action) from non-specific mechanisms that indifferently affect drugs with diverse modes of action. Increased efflux constitutes an important non-specific resistance mechanism. It is based on the overexpression of membrane transporters that pump toxic compounds out of the cell, decreasing intracellular drug concentrations. This mechanism has been widely reported in various organisms such as weeds (Windsor et al., 2003 [4]), bacteria (Li et al., 2015 [5]), cancer cells (Samimi et al., [6]), and fungi [7,8,9]. It is responsible for a cross-resistant phenotype towards several modes of action; hence, it is commonly referred to as multidrug resistance (MDR). In fungi, MDR can rely on the overexpression of two different types of transporters [10], the energy-consuming ATP-binding cassette (ABC) transporters [11], responsible for active drug efflux, or the major facilitator superfamily (MFS) transporters, which use an electrochemical gradient to translocate xenobiotics [12]. As fungal genomes are particularly rich in genes encoding both types of transporters [13], MDR is a frequently encountered resistance mechanism in fungi and yeasts [14,15].

Deciphering the involvement of increased efflux in drug resistance requires the ability to measure the intracellular accumulation of the drug, and to relate this accumulation to the sensitivity of the organism. Therefore, drug efflux studies need to be able to track the fate of the drug once it is applied to the targeted organism, which may trigger several technical difficulties. Indeed, for a long time, efflux studies were based either on the use of radiolabeled or fluorescently labeled drugs, or limited to molecules with intrinsic fluorescence [16,17]. The use of radiolabeled molecules may be cumbersome to set up due to safety measures and regulation. It is, therefore, hard to implement on a daily basis and is now avoided most of the time [16]. Consequently, when possible, the use of fluorescent tags is preferred. However, such labeling requires confirmation that the fluorescent tag does not affect the physical and chemical properties of the drug and, hence, its ability to enter the cell or be pumped out. Regarding drug resistance, this kind of study was, therefore, long restricted to naturally fluorescent molecules and, therefore, severely restricted. Moreover, optical interferences with naturally fluorescent compounds have been reported, especially in studies with plant material, generating errors in the results [18]. In addition, these studies require incubation of the cells in the presence of high concentrations of the drug; thus, it is impossible to assess efflux under standard growth conditions. All these limitations stressed the need to develop a miniaturized and rapid assay suitable for routine use with any unmodified drug.

The technical development of analytical chemistry in recent years, especially the ever-increasing sensitivity of mass spectrometry (MS), has opened the way to a new approach to efflux measurement [16,18]. MS gives access to high-sensitivity quantification in medium to high throughput, even during exponential growth of the studied organisms, and allows quantification of almost any drug regardless of its physical or chemical properties [16,18]. Based on these technical improvements, we designed an HPLC/MS-based assay to follow intracellular fungicide accumulation, monitoring the different levels of fungicide efflux from miniaturized fungal cultures grown with sublethal fungicide concentrations. This is, to our knowledge, the first time such an assay has been used to follow the intracellular accumulation of a fungicide over several days, correlating the intracellular fungicide concentration with the fungicide susceptibility of different fungal strains.

The assay was developed using the wheat fungal pathogen *Zymoseptoria tritici*, a filamentous ascomycete responsible for wheat *Septoria tritici* blotch, the most damaging foliar fungal disease of wheat in western Europe [19,20]. It is a pleomorphic fungus that displays, among other diverse cellular and hyphae morphotypes, a “yeast-like” form, which is easy to handle and to observe under laboratory conditions [21]. In *Z. tritici*, MDR has been well described. It relies mainly on the overexpression of an MFS-type membrane transporter called Mfs1 [8], which is triggered by an insertion in the promotor of the *MFS1* gene [22]. The assay was conducted using the reference strain IPO323 [23] and two isogenic strains, one overexpressing the *MFS1* gene by promotor replacement [22] and the other one displaying a disrupted *mfs1* gene [8]. Our study aimed to quantify the differences in the intracellular fungicide accumulation over a week between these three strains, which were genetically identical except for the *MFS1* locus, therefore displaying contrasting efflux abilities, and to verify whether the measured accumulation can be correlated with the fungicide susceptibility of the strains.

## 2. Materials and Methods

### 2.1. Strains Used in This Study

The IPO323 reference strain was transformed with the plasmid pCAMBIA3100_Mfs1[ZA_HygR_ZB] [8] according to the transformation procedure as described by Zwiers and De Waard, 2001 [24] and the modifications of Kramer et al., 2009 [25] with minor changes. Transformants were selected and isolated on hygromycin-containing medium (100 µg mL^−1^). All transformants were subcultured once on solid YPD medium (Yeast Extract–Peptone–Dextrose: 10 g L^−1^ yeast extract, peptone 20 g L^−1^, 20 g L^−1^ dextrose, 15 g L^−1^ agar) without hygromycin before the final isolation on selective YPD to eliminate unstable transformants.

The remaining transformants were inoculated on solid YPD with hygromycine (100 µg mL^−1^). Eight transformants were checked by PCR (3 min at 95 °C followed by 35 cycles of 15 s at 95 °C, 20 s at 60 °C and 2 min at 70 °C, and a final elongation of 5 min at 72 °C) for *MFS1* gene replacement using the primer-couple QC1_FW + QC1_RV [8], and for *hph* cassette amplification using the primer couple QC2_FW + QC2_RV [8]. Two transformants, #T1 and #T2, produced the expected PCR products (not shown) described by Omrane et al., 2015 [8] and were included in this study.

### 2.2. Culture Conditions

All field strains were grown on solid YPD medium and transformants on YPD medium supplemented with hygromycine (Sigma-Aldrich, Saint-Quentin-Fallavier, France) 100 µg mL^−1^ and incubated at 17 °C under white light for 7 days.

EC_50_ (fungicide concentration leading to 50% growth inhibition in a linear regression model) assays were carried out in 96-well microtiter plates (Sarstedt, Germany) with 10^5^ spores mL^−1^ in 300 µL of liquid YSS (yeast-soluble starch) medium (KH_2_PO_4_ 2 g L^−1^, K_2_HPO_4_ 1.5 g L^−1^, (NH_4_)_2_SO_4_ 1 g L^−1^, MgSO_4_, 7H_2_O 1 g L^−1^, glucose 10 g L^−1^, yeast extract 2 g L^−1^) supplemented with the fungicides in serial dilutions as indicated below. Plates were incubated under light at 17 °C for seven days.

A first assay was performed with carboxin, boscalid, tolnaftate, fludioxonil, and fenpiclonil at 0.0064, 0.032, 0.16, 0.8, 4, and 20 µg mL^−1^ to identify the fungicide displaying the required conditions (impacted by the *MFS1* genotype; EC_50_ value > 0.2 µg mL^−1^). OD_405_ (optical density at λ = 405 nm) was measured at days 0–7 in comparison to sterile medium. A control without fungicide was used to determine the fungicide efficiency.

To determine the boscalid working concentration, the EC_50_ value of all 6 tested strains (Table 1) was determined under the same growth conditions with 6 serial dilutions of boscalid (Sigma-Aldrich, Saint-Quentin-Fallavier, France) in the range of 0.01 to 0.515 µg mL^−1^ with OD_405_ measurement on day 3. Fine adjustments of the EC_50_ values over the 7 days of growth were carried out with 12 dilutions of boscalid in the range of 0.05 to 0.515 µg mL^−1^ in at least 2 biological replicates with 3 technical replicates each.

### 2.3. Preparation of the Analytical Fractions

Cultures were performed in 24-well micro-titer plates (VWR, Rosny-sous-Bois, France). In total, 5 × 10^4^ spores were inoculated in 500 µL of liquid YSS medium supplemented with boscalid under working conditions as indicated in Table 2. Plates were incubated at 17 °C and 100 rpm under light and wells were sampled on days 0, 1, 2, 3, 4, and 7.

Intracellular fractions were prepared as follows (Figure 1). The content of each well was transferred to a 2 mL tube. The well was rinsed with 1 mL of YSS medium and added to the same tube prior to centrifugation at 15,500× *g*, 4 °C for 10 min. The supernatant was discarded, the pellet resuspended in 1 mL of YSS medium, and then centrifuged again under the same conditions. The supernatant was discarded. The pellet was resuspended in 500 µL of acetonitrile (ACN) for LC-MS/MS (Sigma-Aldrich, Saint-Quentin-Fallavier, France) and vortexed before the addition of 500 µL of water (WAT) for LC-MS/MS and approximately 200 mg of glass beads with a 0.5 mm diameter (Dominique Dutscher, Bernolsheim, France) for fast-prep grinding. Grinding was performed in a Fast-prep 24 device (Mp biomedicals, Illkirch, France) at 5.0 m s^−1^ for 50 s. The sample was then centrifuged at 13,000 rpm and 4 °C for 10 min. The supernatant was placed in a new tube and the pellet was resuspended in 500 µL of a 50/50 (*v*/*v*) ACN/WAT mix. The sample was centrifuged under the same conditions as described above and the supernatant was added to the first one. These operations were repeated once. The three pooled supernatants were then analyzed as the “intracellular fraction” (C fraction).

Total fractions were prepared as follows. The content of the well was transferred to a 2 mL tube. The well was rinsed with 500 µL of YSS medium and then added to the previous tube. In total, 500 µL of ACN was added to the tube before vortexing and then glass beads were added for fast-prep grinding. Grinding was performed in a Fast-prep 24 device at 5.0 m s^−1^ for 50 s. The sample was then centrifuged at 13,000 rpm and 4 °C for 10 min. The supernatant was transferred to a new tube and the pellet was resuspended in 250 µL of ACN, vortexed, before the addition of 250 µL of WAT. The sample was then centrifuged under the same conditions as described above and the supernatant was added to the first one. The pellet was then resuspended in 500 µL of a 50/50 (*v*/*v*) ACN/WAT mix, centrifuged under the same conditions, and the supernatant was added to the other two. In total, 500 µL of the supernatant was transferred to a new tube containing 1 mL of WAT for analysis as the “total fraction” (T fraction).

Matrix wells (untreated control) containing 500 µL of YSS medium with 10^5^ spores mL^−1^ were incubated and sampled on days 3 and 7. The fraction was prepared according to the protocol described for the “total fraction” and was designated as the “matrix fraction” (M fraction).

The fungicide control wells contained 500 µL of YSS medium at the appropriate boscalid concentration without *Z. tritici* spores and were incubated and sampled under the same conditions as the total fraction. The whole sample was then analyzed as the “fungicide fraction” (F fraction).

### 2.4. LC-MS/MS Quantification of Boscalid in the Analytical Fractions

Prior to LC-MS/MS analysis, each fraction was diluted in a 3:1 WAT/ACN mix. The C fraction of the ∆*mfs1*::*hph* strain was diluted in a 4:1 WAT/ACN mix to 1/3. In total, 20 µL of each sample was then used for LC-MS/MS injection. Chromatographic separation was performed on an Agilent 1290 Infinity series liquid chromatography system (Agilent technologies, Les Ulis, France). In total, 20 µL was injected through an HTS PAL DLW autosampler (CTC Analytics, Zwingen, Switzerland). A Phenomenex C18 (ODS, Octadecyl), 4 mm L × 2.0 mm i.d. precolumn and a Kinetex C18 100 Å, 2.6 µm, 100 mm × 2.1 mm i.d. analytical column (Phenomenex, Le Pecq, France) were used and a 6-min program was developed for these analyses using a binary A/B solvent system at a flow rate of 0.4 mL/min (solvent A: water, 0.1% formic acid; solvent B: acetonitrile, 0.1% formic acid): 0–3 min (isocratic 50/50), 3–4.5 min (gradient up to 20/80), and 4.5–6 min (gradient down to 50/50).

MS/MS detection was performed on a triple-quadrupole API 5500 Qtrap tandem mass spectrometer (AB Sciex, Les Ulis, France), equipped with a Turbo IonSpray (Electro Spray Ionization) interface operated in positive ion mode and multiple reaction monitoring (MRM). Unit mass resolution was established and maintained in the mass resolving quadrupoles by maintaining a full width at half-maximum (FWHM) of about 0.7 amu. Optimal collisionally activated dissociation (CAD) conditions for fragmentation of the pseudomolecular ion of the analyte were applied, with nitrogen as the collision gas. Data acquisition was performed with Analyst 1.6.2 software (AB Sciex, Les Ulis, France). Boscalid was detected in positive MRM using Q1/Q3 ion transitions at *m*/*z* 343.0/307.0 amu as the quantifier and 343.0/139.9 amu as the qualifier. All source and MS-MS parameters were set up using the analytical standard reference of boscalid (Sigma-Aldrich, Saint-Quentin-Fallavier, France); typical spectra obtained with boscalid are available at https://mobil.bfr.bund.de/cm/343/beispiel_tunefile.pdf (accessed on 7 June 2022). Method validation included the evaluation of the linearity, matrix effect, interference, and stability.

### 2.5. Calculation of the Relative Percentage of Intracellular Boscalid

Relative % of intracellular boscalid = (m_boscalid_[C fraction]/m_boscalid_[T fraction]/OD_520_[T fraction]) × 100.

## 3. Results

### 3.1. Establishment of the In Vitro Growth Conditions for Efflux Assay with Sublethal Fungicide Concentrations

In order to establish our assay, we first considered the detection threshold of the used LC-MS/MS device at 0.003 µg mL^−1^ and the intracellular fungicide accumulation arbitrarily as 10% of the external fungicide concentration. This implied that the minimal fungicide concentration to be used was at least 0.03 µg mL^−1^. Our second condition was that the fungicide concentration in the medium should only weakly impact fungal growth (sublethal). Under these conditions, it should be possible to follow the intracellular fungicide concentration over a period of several days of growth. We, therefore, tested a range of fungicides used against *Z. tritici* or against other plant pathogenic fungi on the growth behavior of three *Z. tritici* genotypes (Table 1) displaying either normal, reduced, or increased efflux. These strains were generated by gene replacement in the reference strain IPO323 ([8,22]; this study). All fungicides were tested in the same concentration range (5-fold serial dilutions of a 20 µg mL^−1^ starting concentration) in a 96-well plate growth assay over 7 days. Preliminary results excluded the DMIs epoxiconazole, tebuconazole, and prochloraz and the SDHIs bixafen and fluopyram as not respecting the indicated thresholds (EC_50_ values too low). Five molecules led to EC_50_ values that were acceptable for our study. These were the SDHIs carboxin and boscalid, the squalene epoxidase inhibitor tolnaftate, and the phenylpyrroles fludioxonil and fenpiclonil (Appendix A). Increased EC_50_ values in the *MFS1^MDR-TypeI^* genotype strains and reduced EC_50_ values in the ∆*mfs1* genotype strains, respectively, were observed only with boscalid and tolnaftate, corroborating previous results showing that both molecules are affected by Mfs1-mediated efflux [8,22]. We decided to pursue the SDHI boscalid as we experienced solubility issues with tolnaftate at the used concentrations.

We then investigated the EC_50_ values of the three genotypes over seven days of growth, and one additional strain included in the analysis (SE31) using a concentration range better adapted to boscalid (see the experimental procedures). The EC_50_ values ranged from 0.2 (IPO323 ∆*mfs1*) to 0.5 (IPO323) and 2 µg mL^−1^ (IPO323 *MFS1^MDR-TypeI^*) on day 3 (Table 2) but increased over time with increasing fungal biomass With respect to the boscalid efflux assay working concentration, we decided to use five-fold lower concentrations than the EC_50_ values on day 3 as the most acceptable compromise between the detection threshold and the lowest impact on growth.

### 3.2. Efflux Assay Set-Up

All strains to be analyzed were grown (for details, see the experimental procedures) in 24-well plates in liquid growth medium supplemented with boscalid concentrations of EC_50_/5 (Table 2). Then, 1 mL aliquots of culture were harvested at 6 time points (n = 7 each), namely on days 0, 1, 2, 3, 4, and 7. Three aliquots per condition (strain*day) were used to determine the total amount of boscalid (T fraction); four aliquots per condition were used to prepare the cellular fraction (C fraction) to determine the intracellular amount of boscalid. As a control, we prepared 1 mL aliquots (n = 12 each day) of growth medium supplemented with each boscalid concentration but without any fungal strain to determine the stability of boscalid over time in the growth medium (F fraction). We also harvested control cultures without fungicides on day 7 to be used as matrix (M fraction) to evaluate the matrix effect on boscalid detection by LC-MS/MS. To test the matrix effect, we added boscalid at final concentrations of 1 or 0.1 ng mL^−1^, respectively, to the M fractions and determined the recovered concentration by LC-MS/MS (Figure 1).

As we observed recovery rates between 75% and 105%, we considered that none of the strains nor the growth medium affected the boscalid detection (no matrix effect). We also tested the stability of boscalid over time, i.e., over seven days under growth conditions without any fungal cell. The recovery rate of boscalid in the F fractions was close to 100%, meaning that boscalid was stable under our tested growth conditions.

### 3.3. Boscalid Accumulates Differently in mfs1 Mutants

We measured the quantity of boscalid by LC-MS/MS in the intracellular fractions (C fractions) of each sample and in the total fraction (T fraction) of the corresponding samples. We also measured the optical density of the crude T fractions at λ = 520 nm. This wavelength, established after a wavelength scan, should allow measurement of different cellular components (e.g., proteins) and cellular debris, and therefore, reflects the fungal biomass. To calculate the relative intracellular concentration of boscalid, we used the following formula in order to normalize the percentage of intracellular boscalid relative to the fungal biomass (Appendix A):

Relative % of intracellular boscalid = (m_boscalid_[C fraction]/m_boscalid_[T fraction]/OD520) × 100.

The data obtained on days 0–2 were too low to detect differences between the strains. Starting on day 3, differences between the three isogenic strains harboring different *MFS1* alleles were observed and became more obvious from day 4 on (Figure 2). In both strains, IPO323 and IPO323 *MFS1^MDR-type I^*, the relative boscalid accumulation remained stable after day 4. In the *mfs1* deletion mutant, the intracellular accumulation decreased after day 4.

While the strain with increased efflux activity (IPO323 *MFS1^MDR-type I^*) had only barely detectable intracellular boscalid concentrations, the corresponding mutant with reduced efflux activity (IPO323 ∆*mfs1*) showed higher intracellular boscalid concentrations than both the wild-type strain IPO323 and the IPO323 *MFS1^MDR-type I^* mutant (1.5–2.3 and 7–15 times higher, respectively). These results indicate that intracellular accumulation of boscalid depends on the efflux activity mediated by the Mfs1 protein.

### 3.4. Intracellular Boscalid Concentrations Correlate with EC_50_ Values

In parallel to the boscalid measurements, we determined the boscalid EC_50_ values during 7 days of growth for all strains in liquid medium in 96-well plates. The EC_50_ values increased over time, indicating that the more fungal biomass, the more fungicide needed to inhibit growth (Appendix A).

We wanted to know whether the intracellular accumulation of boscalid correlated with the sensitivity levels (EC_50_ values) of the strains. Figure 3 shows the negative correlation between both values after 7 days of growth: the more resistant a strain, the lower the intracellular boscalid concentration. We then included the corresponding values of a fourth strain, SE31, a different *MFS1* wild-type strain. SE31 displays an EC_50_ value that is intermediate between IPO323 and IPO323 ∆*mfs1*. The same holds true for its intracellular boscalid accumulation. These four points follow a negative exponential correlation curve with a high confidence index (R^2^ = 0.996) (Figure 3). After 7 days, both strains ∆*mfs1* accumulated on average 1.5 and 11 times more boscalid and were 6 and 23 times more susceptible than their parental strain IPO323 and both *MFS1*^MDR-Type 1^ strains, respectively. For its part, IPO323 accumulated 7 times more boscalid and was 4 times more susceptible than the *MFS1*^MDR-Type 1^ strains. SE31 was 2 times more susceptible than IPO323 and accumulated 1.1 times more boscalid (for calculation, see Appendix A).

## 4. Discussion

Using *Z. tritici*, a wheat fungal pathogen, and boscalid, an SDHI fungicide, we designed an LC-MS/MS-based miniaturized assay to assess fungicide intracellular accumulation over time, and to correlate it with fungicide susceptibility.

Increased efflux contributes to resistance levels by reducing the intracellular concentration of the drug. Globally, only indirect evidence of this phenomenon has been obtained until now by inhibiting the production of membrane transporters or their function [8,16]. However, no direct correlation between drug susceptibility and the precise amount of intracellular drug has been established. This can be explained by the fact that efflux was mostly assessed under specific conditions (after starvation, in synthetic buffer) [16] and/or because the assessment of most drug concentrations was not easily accessible as outlined above. In an attempt to circumvent these limitations, our study shows that it is possible to follow the intracellular accumulation of boscalid applied at sublethal concentrations in *Z. tritici* over a seven-day time course under miniaturized and standard growth conditions using LC-MS/MS. Moreover, we demonstrated that differences in the intracellular accumulation of boscalid were correlated to differences in the boscalid susceptibility. The strains with the disrupted *MFS1* gene (∆*mfs1*) accumulated more boscalid after seven days and were more susceptible than their parental strain IPO323 and the *MFS1*^MDR-Type1^ strains. For its part, IPO323 accumulated more boscalid and was more susceptible than the *MFS1*^MDR-Type1^ strains, showing a negative correlation between the accumulated amounts of boscalid and the susceptibility.

To establish this correlation, we used isogenic strains, so we can assume that the differences in the boscalid susceptibility were only due to the level of expression of the *MFS1* gene. Using LC-MS/MS, we were able to demonstrate that fungicide accumulation in the cells, which negatively correlates with their efflux capacities, can explain differences in the fungicide susceptibility between isogenic strains that only differ by their level of drug efflux.

In addition, to validate the correlation, we conducted the same assay using a genetically unrelated strain. For this strain, called SE31, the correlation suited the correlation curve. This result implies that if the correlation remains true for an unknown strain, its susceptibility to boscalid is due to differences in its efflux ability (Figure 3). On the contrary, a strain for which the data deviates from the correlation would see its susceptibility explained by factors other than different efflux abilities only (e.g., target modification).

Limitations and perspectives of this study. Our study showed that the increasing sensitivity of analytical methods, especially LC-MS/MS, allows in-depth study of intracellular drug accumulation. However, with increasing efflux abilities, the amount of intracellular drugs may be very low. Here, to circumvent this problem and to set up the assay, we used boscalid, which is a fungicide with only weak activity against *Z. tritici*. It is likely that the detection of intracellular concentrations of more active drugs may remain difficult, even for highly sensitive LC-MS/MS systems. In such situations, the possibility of reaching detectable concentrations could be achieved through the use of larger volumes and, hence, more biomass. However, an increase in the culture volumes means a reduction in the assay’s throughput. Therefore, the parameters of the experiment constitute a compromise between the miniaturization of the assay and the detection thresholds of LC-MS/MS devices.

Except for its sensitivity, LC-MS/MS also comes with its own technical difficulties. Each component of the samples that is not a solvent or assessed compound (i.e., components from the growth medium or cellular debris for example) may modify the intensity of the output signal. This phenomenon, known as the “matrix effect”, can lead to an increased or decreased signal intensity. Given that it modifies the signal, the matrix effect also affects the quantification, implying the implementation of specific control samples to assess the nature and intensity of the effect. The matrix effect may also increase the variability of this assay due to the fungal biology and the numerous steps of sample preparation.

In conclusion, using a miniaturized LC-MS/MS-based efflux assay, we were able to demonstrate that fungicide accumulation in cells can explain differences in the fungicide susceptibility between isogenic strains that only differ by their level of drug efflux. Indeed, intracellular fungicide accumulation negatively correlates with efflux efficacy. In addition, the established correlation curve makes it possible to correlate fungicide susceptibility and efflux in unknown strains. This assay may be useful in lead development when a new molecule that is highly active against its isolated target protein displays only weak fungicidal activity against the target organism during in vitro growth. Different mechanisms may hinder fungicidal activity (e.g., metabolization, reduced influx, increased efflux). With the above-described method, it will be possible to screen such compounds in medium to high throughput and to identify whether the observed loss of activity is due to increased efflux.

## Figures and Tables

**Figure 1 microorganisms-10-01494-f001:**
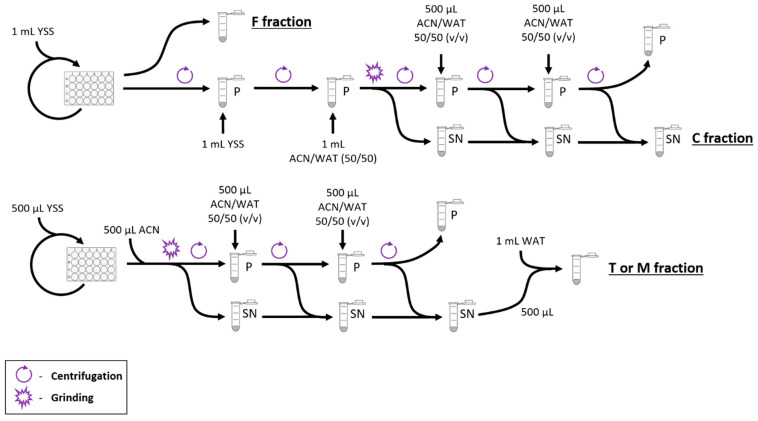
Flowchart of the fraction preparation. F: fungicide fraction; C: intracellular fraction; T: total fraction; M: matrix fraction. YSS: YSS liquid medium; ACN: acetonitrile; WAT: water; P: pellet; SN: supernatant.

**Figure 2 microorganisms-10-01494-f002:**
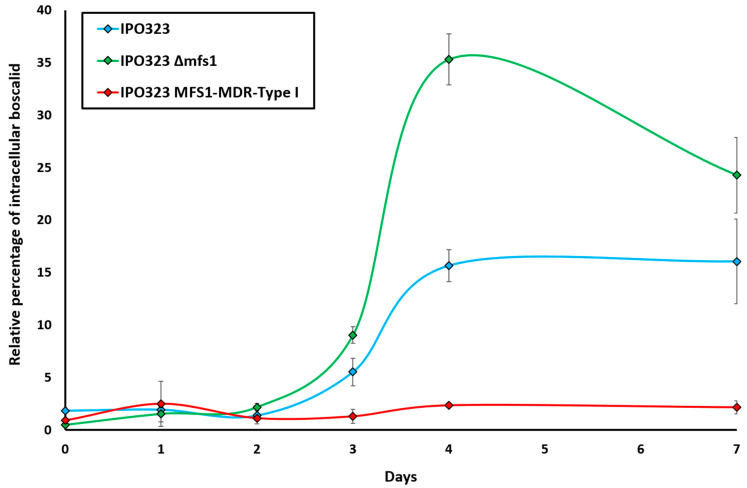
Intracellular accumulation of boscalid in *Z. tritici mfs1* mutants during in vitro growth. Indicated values are the means of n = 3 technical replicates of IPO323, and of two independent mutants per *MFS1* genotype.

**Figure 3 microorganisms-10-01494-f003:**
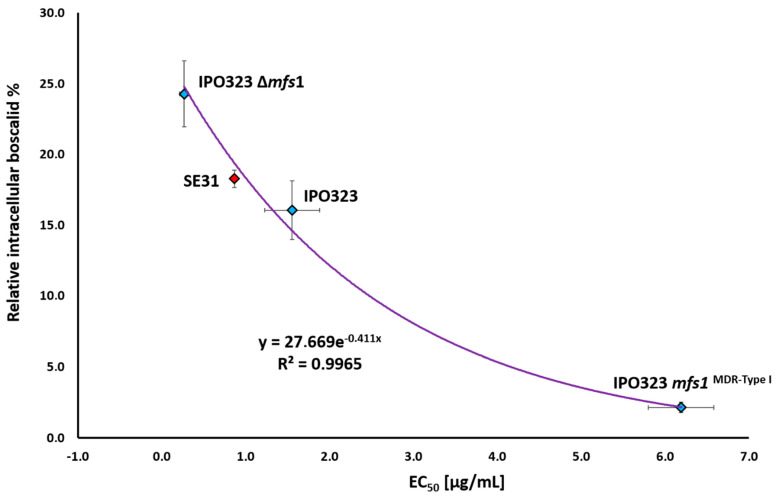
Exponential correlation curve between the intracellular accumulation of boscalid (day 7) and EC_50_ values (day 7) of *Z. tritici* strains with different *MFS1* genotypes. The randomly analyzed fourth strain SE31 is highlighted in red. The EC_50_ values are the means of 2–4 biological replicates. The intracellular boscalid percentages are the means of three independent measures, including two biological replicates, and the means of two independent mutants per *MFS1* genotype. Error bars = standard errors.

**Table 1 microorganisms-10-01494-t001:** Origins and genotypes of the *Z. tritici* strains used in this study.

Strain	*MFS1* Genotype	Origin, Reference
IPO323	WT	Field [23]
SE31	WT	Field (Bayer internal reference)
IPO323 Δ*mfs1* #T1 IPO323 Δ*mfs1* #T2	∆*mfs1*::*hph* with disrupted *MFS1* gene	IPO323 transformant [8] and this study
IPO323 *MFS1*^MDR-Type I^ #6.18 IPO323 *MFS1*^MDR-Type I^ #6.20	*MFS1*^MDR-Type I^ with replaced *MFS1* gene	IPO323 transformant [22]

IPO323 ∆*mfs1*—construction and validation.

**Table 2 microorganisms-10-01494-t002:** Boscalid EC_50_ and working concentration determination. EC_50_ values were determined using a linear regression model of the dose–response curves on day 3. N = 4 (SE31) to 15 (IPO323 Δ*mfs1*).

Strain	EC_50_ Average (µg mL^−1^)	Working Concentration Efflux Assay (µg mL^−1^)
IPO323 Δ*mfs1* *	0.185 ± 0.043	0.036
IPO323	0.519 ± 0.128	0.100
SE31	0.498 ± 0.051	0.100
IPO323 *MFS1*^MDR-Type I^ *	2.098 ± 0.857	0.420

* indicated values are the means of *n* = 2–4 biological replicates of 2 independent mutants per genotype.

## Data Availability

Not applicable.

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
