# Peer review of "LC-MS/MS-Based Fungicide Accumulation Assay to Demonstrate Efflux Activity in the Wheat Pathogen Zymoseptoria tritici"

_microorganisms, 2022, doi:10.3390/microorganisms10081494_

Round 1

Reviewer 1 Report

The study was carried out to detect the intracellular accumulation of a fungicide in the wheat pathogen Zymoseptoria tritici over several days by using LC-MS/MS, to verify if measured accumulation can be correlated to the fungicide susceptibility of the strains. This may be the first time to use this method to track the intracellular accumulation of fungicides within a few days, and correlate the intracellular fungicide concentration with the fungicide sensitivity of different fungal strains, so it is worth studying.

There are some comments as follows.

  1. In the part of Materials and Methods, there is a lack of statistical analysis. In the result part, if the difference analysis is involved (such as P214), the significance level needs to be pointed out.
  2. The citation format in the text are inconsistent, or appeared in the form of serial number, or in the form of "author + year of publication" (such as Table 1), which should be unified according to the requirements of journals.
  3. The use of units is chaotic, which is specifically reflected in the volume units L and l, µg.ml-1 and µg / ml. In addition, remove the space between the value and % (P244, p248, etc.).
  4. The semicolon should be represented by a comma, not a period(P137,145,154).
  5. In the Material and Method part of the article, the multiple sign should be in the form of symbol rather than the English letter X.

Author Response

There are some comments as follows.

1. In the part of Materials and Methods, there is a lack of statistical analysis. In the result part, if the difference analysis is involved (such as P214), the significance level needs to be pointed out.

In fact, no statistics were applied for the comparaison of the EC50 values of Table S1. The sentence was reformulated and the word “significant” withdrawn.

2. The citation format in the text are inconsistent, or appeared in the form of serial number, or in the form of "author + year of publication" (such as Table 1), which should be unified according to the requirements of journals.

Done

3. The use of units is chaotic, which is specifically reflected in the volume units L and l, µg.ml-1 and µg / ml. In addition, remove the space between the value and % (P244, p248, etc.).

The units were homogenized.

4. The semicolon should be represented by a comma, not a period(P137,145,154).

Done

5. In the Material and Method part of the article, the multiple sign should be in the form of symbol rather than the English letter X.

Done

Reviewer 2 Report

This manuscript entitled “LC-MS/MS based fungicide accumulation assay to demonstrate efflux activity in the wheat pathogen Zymoseptoria tritici” aimed to develop an LC/MS-based assay to monitor intracellular fungicide accumulation, as an indicator of pump efflux activity. Some major points need improvements and better explanation before it can be suggested for publication.

Majors

  1. In section 3.3, the absorbance of the crude T-fraction was used for normalization, why 520 nm? Do those biological molecules, such as proteins and nucleic acids, have absorbance at this wavelength.

  1. Line 244, “As we observed recovery rates between 75 and 105 %, we considered that none of the strains neither the growth medium affected boscalid detection.” How to calculate the recovery rates? How did the authors make such a conclusion of not affecting boscalid detection based on the various recovery rates.

  1. Different fractions, C and M, went through different preparation procedures, and would they have different recovery rates? Would the recovery rates variations cause any bias on the relative intracellular boscalid accumulation?

  1. What would possibly happen if other drugs to be used for detection? Is this experimental approach suitable for application of the intracellular accumulation of other drugs?

  1. As we know that salt interferences might cause signal suppression, how did the author prevent this happened in this study?

  1. Did the authors calculate the concentration of boscalid from othe peak areas in LC-MS/MS? By the way, the MS spectrum for boscalid should be added.

  1. In figure 3, the EC50 and relative drug accumulation for the strains were correlated.

  1. LC-MS and MALDI-TOF spectrometry have been used to investigate the efflux pump efficiencies of drug-resistant microorganisms, by directly monitoring the intracellular or extracellular drug concentrations. The authors are encouraged to include more relevant references for discussions.

Minors

  1. The authors were encouraged to discuss the various EC50 data for different drugs in Table S1.
  2. Unit should be added to the table spanner in Table S2.
  3. Some words should be italicized, such as m/z and tritici.
  4. The formula of boscalid accumulation can be added into M&M.
  5. In Figure 2, why curves instead straight lines were used?
  6. Were there any technical difficulties when you were dealing with fungal cells rather than bacterial cells?

Reviewer 3 Report

In this article, the authors describe a throughput method to assess intracellular drug concentration in the plant pathogenic fungus Zymoseptoria tritici. In addition, they have tried to correlate the drug efflux with the susceptibility to boscalid in the tested strains. Using isogenic mutant strains with differing efflux activity, the authors could validate the negative correlation between intracellular boscalid concentration and efflux activity. The study is promising; however, the authors need to address the comments below and modify the manuscript accordingly.

L35: Among non-specific mechanisms figures increases efflux à Re-write the sentence.

L38, L39: e.g. à elaborate

L93: Z. tritici à Zymoseptoria tritici

L97: by 24 and by 25 à need to write name of the first author followed by et al. Please correct it similar writings throughout the manuscript

L133: 17°C to 17 °C

L135: cf. Fig. 1

L137: 15,500 g

L199: 0.003 μg.ml-1 à 0.003 μgml-1

L351-354: It is not very clear. Need to rewrite the sentence

L356-360: It is not clear. Need to rewrite the sentence

The authors mention ‘matrix effect’ as one of the drawbacks of LC-MS/MS. Are there any suggestions to reduce the matrix effect in this study especially when dealing with miniature assays?

L362: in vitro à italics

Round 2

Reviewer 2 Report

1.    Sample preparation procedure is important for a method paper, but it is a bit confusing to understand the sample preparation in M&M 2.3 and figure 1.

(1)  It seems that the ACN/WAT mix extraction is accomplished for sample preparation (C fraction)? It should be modified for better clarity.

(2)  In addition, why ACN/WAT mix instead of methanol/water mixtures? Can ACN/WAT mix extract most hydrophilic and hydrophobic molecules? Because this is a method aiming to analyze various drugs.

2.    This is a paper to detect boscalid using LC MS/MS, the authors should include a boscalid MS spectrum in the manuscript.

3.    Previous studies using MS to monitor drug accumulation compare their data from MS with some conventional studies, such as fluorescent studies, for validation. The authors aim to provide a standard procedure to quantify the intracellular boscalid in fungal cells. How to know if the preparation/detection procedures are reliable if there was no comparison for validation.